# Single-Equipment with Multiple-Application for an Automated Robot-Car Control System

**DOI:** 10.3390/s19030662

**Published:** 2019-02-06

**Authors:** Saleem Ullah, Zain Mumtaz, Shuo Liu, Mohammad Abubaqr, Athar Mahboob, Hamza Ahmad Madni

**Affiliations:** 1Department of Computer Science, Khwaja Fareed University of Engineering and Information Technology, Rahim Yar Khan 64200, Pakistan; saleem.ullah@kfueit.edu.pk (S.U.); zainmumtaz007@gmail.com (Z.M.); mohammadabubaqr@outlook.com (M.A.); athar@kfueit.edu.pk (A.M.); 2State Key Laboratory of Millimeter Waves, Department of Radio Engineering, Southeast University, Nanjing 210096, China; liushuo.china@seu.edu.cn; 3Department of Computer Engineering, Khwaja Fareed University of Engineering and Information Technology, Rahim Yar Khan 64200, Pakistan

**Keywords:** android, arduino, bluetooth, hand-gesture recognition, low cost, open source, sensors, smart cars, speech recognition

## Abstract

The integration of greater functionalities into vehicles increases the complexity of car-controlling. Many research efforts are dedicated to designing car-controlling systems that allow users to instruct the car just to show it what it should do; however, for non-expert users, controlling the car with a remote or a switch is complicated. So, keeping this in mind, this paper presents an Arduino based car-controlling system that no longer requires manual control of the cars. Two main contributions are presented in this work. Firstly, we show that the car can be controlled with hand-gestures, according to the movement and position of the hand. The hand-gesture system works with an Arduino Nano, accelerometer, and radio-frequency (RF) transmitter. The accelerometer (attached with the hand-glove) senses the acceleration forces that are produced by the hand movement, and it will transfer the data to the Arduino Nano that is placed on hand glove. After receiving the data, Arduino Nano will convert it into different angle values in ranges of 0–450° and send the data to the RF receiver of the Arduino Uno, which is placed on the car through the RF transmitter. Secondly, the proposed car system is to be controlled by an android based mobile-application with different modes (e.g., touch buttons mode, voice recognition mode). The mobile-application system is the extension of the hand-gesture system with the addition of Bluetooth module. In this case, whenever the user presses any of the touch buttons in the application, and/or gives voice commands, the corresponding signal is sent to the Arduino Uno. After receiving the signal, Arduino will check this against its predefined instructions for moving forward, backward, left, right, and brake; then it will send the command to the motor module to move the car in the corresponding direction. In addition, an automatic obstacle detection system is introduced to improve the safety measurements to avoid any hazards with the help of sensors placed at the front of the car. The proposed systems are designed as a lab-scale prototype to experimentally validate the efficiency, accuracy, and affordability of the systems. The experimental results prove that the proposed work has all in one capability (hand-gesture, touch buttons and voice-recognition with mobile-application, obstacle detection), is very easy to use, and can be easily assembled in a simple hardware circuit. We remark that the proposed systems can be implemented under real conditions at large-scale in the future, which will be useful in automobiles and robotics applications.

## 1. Introduction

A robot is an intelligent machine controlled by a computer application to perform various operations and services. Robots play an essential role in automation across all sectors like manufacturing, military, construction, and medical [1] etc. Robots can not only help humans save time but also increase productivity [2], efficiency, reliability, reduce the use of resources to save energy, and reduce the running cost etc. Meanwhile, robots play a significant role in providing help in such tasks that cannot be done smoothly by a disabled person, i.e., controlling a car using physical devices has become very successful. There are two categories of robots: autonomous robots (edge sensing robots [3], line sensing [4]) and remote-controlled robots (gesture-controlled robots [5]). Therefore, the employment of gesture-controlled robots is one of the more elegant and aesthetic options termed to catch the human-gesture, which is difficult to understand in machine-learning.

There are many methods to capture gesture. Commonly used methods include a data glove [6], camera [7], infrared waves [8], tactile [9], acoustic [10] and motion technology [11]. These embedded systems [6,7,8,9,10,11] are designed for particular control and can be optimized to increase reliability, performance, and reduce the cost and size of the device. Moreover, researchers are showing tremendous interest in gesture recognition, building robots and many other devices that are directly controlled by human gestures. The gesture control mechanism is applied in various fields like socially assistive robots, augmented reality, emotion detection from facial expressions, and recognition of sign languages [12,13,14] etc. Furthermore, the emotional gesture [12] identification from the face is also under investigation. Similarly, with the arrival of the smartphone and other advanced technologies, operating machines have become more adaptable. In this regard, the smartphone communicates with the microcontroller of the robot via radio frequency (RF), Bluetooth or Wi-Fi.

The traditional robot system has been limited to the remote system in which the desired actions can only be done with a unique and specific remote. In this scenario, if the remote is broken or lost, the robot loses control and leads to hazards and is waste of money as well. To overcome this remote-control concept, controlling the robot using a gesture recognition technique with an accelerometer [15,16,17,18,19,20,21,22,23,24,25,26,27,28,29,30,31,32,33], Bluetooth android application [34,35,36,37,38,39,40,41], and voice recognition [42,43,44,45,46,47,48,49,50] have been proposed. Thus, in previous literature, most of them are line-sensing robots [51,52,53] that are controlled with Infrared LEDs, HF band short-range RFID systems [54], and CNY 70 sensors [55]. In addition, vision-based technology [56,57] is also used to control the movement of the robot. Furthermore, an automated robot control system with a six sense [58,59,60] and ZigBee based system [61,62,63,64] has also been implemented. In addition, another approach has been introduced to allow the robot to avoid obstacles [65,66,67], in which the robot stops moving or changes its route of motion. Similarly, in edge detector [3] systems, the robot keeps moving due to the detection of an edge; and in an line follower systems [4], the robot follows a unique line and starts moving. A motion capture sensor recognition system [16,22] is also used to implement an accelerometer-based method to interact wirelessly with an industrial robotic arm.

Apart from the traditional robot control system, the term “internet of things” (IoT) [68,69,70,71,72,73,74] is also essential for connecting a robot with the internet to allow users to control the robot from anywhere at anytime. These wireless systems are providing vital help to robot self-regulation systems by using Wi-Fi and cloud computing, etc. As far as we know, a need still exists for the design of single-equipment with a multiple-application system that supports various tasks (e.g., hand-gesture recognition, android mobile application control and voice recognition concepts, monitoring the obstacles) and is very easy to use and can be easily assembled in a simple hardware circuit.

In this paper, we introduce the design and experimentally demonstrate that a robot-car can be controlled with hand movement by using the technique of hand-gesture. This work is accomplished with the conventional arrangements of the Arduino microcontroller, accelerometer, RF transmitter/receiver, Android mobile-application, Bluetooth, and motor module. The proposed robot-car is controlled via gesture recognition technique by measuring the angles and position of the hand. In this way, the robot will move according to the movement of the hand. Furthermore, we extend this system to allow the robot to be controlled by just a click on the cellphone with an Android operating system, and voice recognition via Bluetooth technology. There are two main controllers in android the application. The first one is the touch buttons. The robot will move accordingly as the user touches the button. The second one is voice recognition. The robot will follow and move accordingly as the user says the operating command. Most importantly, an obstacle detection sensor is set to detect the obstacle in front of it, and when a sensor detects the obstacle, it stops moving. Hence, the proposed systems of controlling the robot car with both gesture control and android application control are performed and displayed in a lab-scale prototype to confirm that the proposed designs can be easily implemented in large and real-scale conditions in the future.

The innovative element of the proposed work in this paper is an Arduino that is a user-friendly microcontroller that can be easily available on demand. On the other hand, automated systems can be designed with raspberry pie, ZigBee, and other microcontrollers that are costly and complicated for the encapsulation process to assemble the various functionalities in a simple hardware circuit. An accelerometer based hand-glove is also the key element to investigate the car-controlling set-up as a reciprocal of vision-based gesture recognition [75] (the aim is in the field of service robotics to control a robot equipped with a manipulator), and finger-gesture recognition [76] (the aim is to make it feasible to interact with a portable device or a computer through the recognition of finger gestures). 

Furthermore, the motivation of conducting this research is to facilitate the old-age and/or physically disabled people who cannot walk or who are facing problems in performing their daily life activities. The proposed work in this paper can be easily implemented to design and operate a wheelchair with the help of hand-gestures and mobile-application systems, which in turn commands the wheelchair. As the wheelchair moves front, back, right, and left, a partially paralyzed patient can move freely. Most importantly, the person will be able to control the wheelchair in a narrow space because this proposed scheme uses sensors that are helpful in avoiding collisions. Similarly, it might be difficult or unsafe for humans to do some tasks like picking up dangerous chemicals. Thus repeated pick and place actions are important in certain industries. In this field it is quite complicated to control the robot or machine with a remote or switches; sometime the operator may get confused between the switches and buttons, so we aim to design a new concept to control the machine by hand movement combined with a Bluetooth android application that will simultaneously control the movement of the robot. Thus, the aim of this proposed research is to construct a multiple-functional robot with low cost that can be used in various activities. 

The remaining content of the paper is ordered as follows. In Section 2, the idea of the automatic control robot car is introduced with a detailed explanation of the electronic components that are used in the proposed system, based on gesture recognition with the mobile-based android application. Also, the experimental results of the proposed systems in a lab-scale prototype are presented in Section 3. Section 4 concludes the paper. Section 5 presents the future work. 

## 2. Materials and Methods

For the simplicity of analysis, Figure 1 demonstrates the complete working mechanism and the features of the proposed automatic robot car. Whereas, I/P and O/P represent the flow of the system as input and output. There are two modes of transmission and controlling of the robot car. The first one is the hand-gesture system. The hand accelerometer first senses the acceleration forces from the direction of the hand and sends them to the Arduino Nano that is attached to the hand. After receiving the data, Arduino Nano converts it into different angles, between 0–450°, and sends it to the RF receiver of the Arduino Uno that is attached to the robot car through the RF sender.

After receiving the data, the Arduino Uno of the car will measure the received angles with the predefined set of angles and send a signal to the motor module to move the wheels of the robot car accordingly to the angles. It can be noticed that the range of angles are predefined for the wheel movement of the robot car to move forward, backward, brake, left and right.

The second mode is to control a robot car with an android mobile application, which is a built in android mobile application available at the google play store that can be easily downloaded [77]. In this system, when the user presses the corresponding touch button, a signal is transferred to the Arduino UNO that is attached to the car through the built-in mobile Bluetooth device. After receiving the following signal command, Arduino will check this signal with a predefined instruction that is programmed via coding and send the following signal to the motor module to move the wheels of the robot car accordingly to the received signal.

### 2.1. Electronic Components

Various electronic components are used for creating electronic circuits. Consequently, our proposed circuit diagrams also contain those components that are specified in Table 1.

#### 2.1.1. Arduino UNO

The Arduino Uno [50,78] is a microcontroller board that is principally based on the ATmega328 microcontroller series and has an integrated development environment (IDE) to write, compile, and upload the programming codes to the microcontroller. Various sensors will forward the environmental data as an input to the microcontroller, which will correspondingly send the attached peripherals, e.g., actuators. It has a total number of 28 pins; 14 digital input/output pins (six are pulse width modulation (PWM) pins) and six pins are the analogs used for interaction with the electronic components like sensors, motors; 3 GND pins (for grounding) and remaining pins for 3.3 V, 5 V, VIN, RESET and AREF (Analogue Reference). Arduino contains a microcontroller with a 32 KB storage memory, 2 KB of SRAM (Static random-access memory), and 1 KB of EEPROM (Electrically Erasable Programmable Read-Only Memory). Arduino primarily supports a C programming language compiler, macro-assemblers, and evaluation kits. It also has 16 MHz ceramic resonators, a USB connection jack for connecting with a computer, a jack for external power supply, an ICSP (in-circuit serial programmer) header, and a reset button to reset to the factory settings. It operates with an input voltage of 7 to 12 V (limit up to 20 V).

#### 2.1.2. Arduino Nano

Arduino [79] is comprised of a microcontroller of ATmega328P with 32 KB memory, of which 2 KB is used for the bootloader, 2 KB of Static random-access memory (SRAM), 16 MHZ Clock Speed, and 1 KB of electrically erasable programmable read only memory (EEPROM). ATmega328P supports C compiler, evaluation kits, and macro-assemblers. It has a total 30 pins of which 14 are digital input and output pins (six are pulse width modulation (PWM)), 8 Analog IN pins, 2 GND pins, 2 RESET and remaining pins for VIN, 5V, 3V and REF respectively. It has a USB connection jack, an external power supply jack, an ICSP header and a reset button. It operates with an input voltage of 7 to 12 V (limit up to 20 V).

#### 2.1.3. MPU-6050 Accelerometer

The MPU-6050 sensor [80] holds a MEMS (Micro-Electro-Mechanical Systems) accelerometer and a MEMS gyroscope only in the single chip. It contains 16-bits analogue to digital conversion hardware for each of the channels and it captures the x, y, and z channel at the same time. MPU-6050 interfaces with the Arduino microcontroller using the I2C protocol. It has a total number of eight pins named as VCC (for provides power), ground (for grounding of system), serial clock (SCL used for providing clock pulse for I2C communication), serial data (SDA used for transferring data through I2C communication), auxiliary serial data (XDA is optional but can be used to interface other I2C modules with MPU6050), auxiliary serial clock (XCL is also optional but can be used to interface other I2C modules with MPU6050), AD0 (If more than one MPU6050 is used a single MCU, then this pin can be used to vary the address). Interrupt pin (INT) indicates that data is available for MCU to read, and operating voltage is 3–5 V.

#### 2.1.4. RF Transmitter and Receiver

RF modules are 433 MHz RF transmitter and receiver modules [81] that are used to transmit and receive the infrared waves. RF transmitter consists of three pins; ATAD (Signal pin used to send data for the receiver), VCC (for providing voltage) and Ground pin (for grounding the module). Its working voltage is 3–12 V and consumes 12 V power. The RF transmitter can transmit up to 90 m in an open area. The RF Receiver module consists of four pins; VCC (for providing Voltage), two DATA pins for receiving data from transmitter and Ground (for grounding the module). Its working voltage is 5 VDC.

#### 2.1.5. L293D

The L293D [82] is quadruple high-current half-H drivers that are used to control the direction and speed of up to four direct current (DC) motors with individual 8-bit speed selection simultaneously up to 0.6 A each. It is designed to provide bidirectional drive currents of up to 600-mA at voltages from 4.5 to 36 V. It has eight output pins for connecting four DC motors or two Stepper motors, one reset button, six pins for connecting two servo motors, one +M pin for providing external motor power, five Analog input pins, thirteen digital input pins for connecting with Arduino and ground pins for grounding.

#### 2.1.6. Bluetooth Module HC-05

The HC-05 [83] is a Bluetooth SPP (Serial Port Protocol) module that is designed for direct wireless serial connection setup and can be used in a Slave or Master configuration, giving it an outstanding solution for wireless transmission. This serial port Bluetooth module is entirely adequate Bluetooth V2.0 + EDR (Enhanced Data Rate) 3 Mbps Modulation with complete 2.4 GHz radio transceiver and baseband. It has a total number of six pins; an ENABLE pin to toggle between the Data Mode and AT command mode, a VCC pin for providing voltage, a Ground pin for grounding, a TX-Transmitter for transmitting serial data, a RX-Receiver for receiving the serial data, and a State pin (to check if Bluetooth is working correctly or paired/unpaired). Its input voltage is 3.3–5 V and can transmit up to 90 m.

#### 2.1.7. Android Mobile Application

An Android application is a software application (developed with a computer programming language) which run on the Android platform. The application for controlling a robot car is available at [77] and easily downloadable.

#### 2.1.8. L298N Motor Module

An L298N motor module [84] is a heavy-duty dual H-bridge controller, which is used to control the direction and speed of single or two direct current (DC) motors of up to 2 A each, having a voltage between 5 V to 35 V. It has principally four output pins for the connection of the DC motors, four input pins to receive the signal from the microcontroller, two enable jumpers (remove one of the corresponding jumpers and connect to the pulse width modulation pins to control the speed of DC motors). It also has an onboard 5 V regulator that removes that regulator if the supply voltage is up to 12 V.

## 3. Designing Methodology

### 3.1. Hand-Gestures Recognition

Figure 2 shows the circuit design of hand-gestures, which control the robotic car using hand movement. In this scenario, the robotic car will move in the same direction as the direction of hand rotation. Figure 2a is for the hand-gesture setup and Figure 2b is for the car setup. First, we describe the hardware implementation of the hand-gesture setup. In this task, one Arduino Nano, one MPU-6050 accelerometer and one RF transmitter were used. The SLC pin of the MPU-6050 module is connected to the Arduino analogue Pin A5, SDA to the A4, Ground pin to the GND and VCC (voltage at the common collector) pin to the 5 V pin of the Arduino. DATA pin of the RX-transmitter is connected to the Arduino digital PIN D8, VCC pin to 5 V and ground pin to GND port, as shown in Figure 2a. 

After hardware implementing of the hand gesture, we move to the implementation of car setup. In this task, one L293D motor module, one Arduino Uno, and one RF-receiver were used. Digital pins from A0–A13 of L293D motor module are connected to the digital pins A0–A13 and Analog pins A0–A5 are connected to the analog pins A0–A5, and similarly, Vin pin to Vin port, 3 V pin to 3 V port, 5 V pin to 5 V port, reset pin to reset port, ground pin to GND port and AREF pin to the AREF port of Arduino Uno as displayed in Figure 2b. Further, M1.1–M4.1 pins are connected to the negative terminal and M1.2–M4.2 pins of the motor module are connected to the positive terminals of the motor. Similarly, a VCC pin of the RF-receiver is connected to the VCC pin of the motor module, the DATA pin to the digital pin D2 of the motor module and Arduino and at last, the Ground pin to the GND port of the Arduino Uno. The complete software code of this case is presented in Appendix A. 

#### 3.1.1. Movement of Motors with Hand-Gesture

As the user moves their hand, the reading of the accelerometer will change, and then it will be recaptured by the application. There are genuinely two values: One is the minimum value (X_range_), and another is the maximum value (Y_range_), and the range is defined using these two values for each function of the car. In simple words, the set of ranges are defined for the movement of the robot-car in a specific way. If the received data by the application lies within these specified values, then the corresponding decision will be made. This decision value will be sent to the microcontroller, which then processes it to understand the corresponding gesture. After knowing the gesture, it will send a signal to move the robot car accordingly. There is a total of four DC motors, one motor for each wheel (two for left diagonal wheels, and two for right diagonal wheels) used in the construction of this car. The motors are controlled by the L293D motor shield.

Figure 3 represents the main idea of gesture recognition and car movement. When the user tilts his hand downward, the gesture is recognized as the forward movement, all four wheels of the motors will rotate forward and the robot will move in the forward direction, which can be easily seen in Figure 3a. Figure 3b illustrates the case, when the user tilts his hand upwards, the gesture is recognized as the move backward gesture, and all the four wheels of motors will rotate backward, and the robot moves in a backward direction. In Figure 3c, the hand is in the straight position, and if the gesture is recognized to stop the car, all four wheels will stop moving. When the user tilts his hand in the right direction, the gesture is recognized as the right turn, so the right diagonal motors (top right and back right motors) will rotate forwards, and the robot moves in the right direction as shown in Figure 3d. Similarly, when the user tilts his hand in the left direction, the gesture is recognized as the left turn, so only left diagonal motors (top left and back left motors) will rotate forward and the robot moves in the left direction, as represented in Figure 3e.

It is worth noting that the values of the tilt of the user’s hand determine the right or left turn is a cutting turn. The cutting turn is the one in which the car changes its direction without slowing down speed. If it is a right turn, the car will start moving in the right circle direction, or if it is left turn, the car will move in the left circle direction. The hand of a user will never lie within the two gestures (the values for each action is defined differently from another action), i.e., the value of the accelerometer will not lie in the threshold values of two directions (left turn and forward, left turn and backward, right turn and forward, right turn and backward).

#### 3.1.2. Results and Discussions

In the beginning, the accelerometer (attached with the hand glove) will sense the acceleration forces that are produced with the hand movement at that time, and it will consequently transfer the data to the Arduino Nano that is placed on the hand glove. After receiving the data, the Arduino Nano will convert it into different angle values of ranges 0–450° and send the data to the RF Receiver of the Arduino Uno that is placed on the car through the RF Transmitter, as is clearly exhibited in Figure 4. 

When Arduino Uno receives these angles, it will measure these with the predefined set of angle values. Arduino Uno will check if X_range_ lies between 325–345° and Y_range_ lies between 380–410°, it will send a forward signal to motor shield. If Arduino Uno founds X_range_ in between 315–345° and Y_range_ in between 250–275°, it will send a backward signal to the motor shield. Similarly, if X_range_ lies between 385–415° and Y_range_ between 315–350°, the Arduino will send a Left signal to the motor shield. Furthermore, if X_range_ lies within 255–275° and Y_range_ lies within 320–345°, Arduino will send a right signal to the motor shield. Thus, after the obtained signal, the motor shield will control the car’s movement. 

Initially, the robot car will not move and stay motionless when it does not receive any signal from an accelerometer or when the accelerometer of hand glove is in the straight position. Whenever the user tilts or rotates his hand palm glove to any direction, the angles will be measured of that direction by an Arduino and a command will be sent to the motor shield to turn ON the corresponding motors and the robot car will start moving in a similar to the hand palm direction. 

Figure 5 shows the final presentation of the proposed robot car system that controlled it via a hand gesture recognition using Arduino. Figure 5a represents the complete hardware implementation of the hand glove and the robot car. Figure 5b shows that the robot car is moving to the forward direction (all the four wheels is moving) because the hand is bent in the downward position. In Figure 5c, the robot car is moving in a backward direction as the hand’s position is upwards. In Figure 5d, the robot car is not moving because the hand is in the straight position as a brake command. In Figure 5e, the robot car is moving in the right direction because the hand is turned in the right position. Similarly, in Figure 5f, the car is moving to the left direction because the hand is turned in the left position.

### 3.2. Mobile Application System

According to the motive, the primary idea of this paper is to produce an innovation for the robot car that will allow it to be controlled by hand-gestures as well as an Android mobile application. As presented in Figure 4, the robot car is controlled via hand-gestures. Here, we enhanced our task by controlling the robot car based on the Android mobile application (e.g., touch button control and voice recognition), and the circuit design of this mode in a lab-scale model can be seen in Figure 6. 

In this task, an Arduino UNO, one L298 motor module, one HC-05 Bluetooth module, one 12 V battery and four DC motors are used. Hence, the TX, RX pin of the Bluetooth module is connected to pin number D0, D1, VCC pin to 5 V port and ground pin to ground port of the Arduino UNO. Similarly, the 5 V pin and ground pin of the motor module is connected to 5 V port and ground port of Arduino UNO. The 12 V pin of the motor module is connected to a positive terminal of a 12 V battery and the negative terminal of the battery is connected to the ground port of Arduino, motor module, and Bluetooth module.

Moreover, IN1–IN4 pin of the motor module is attached to Arduino PIN D9–D12 respectively. OUT1 pin of the motor module is connected to the positive terminal of the LWF motor (left wheel forward) and the negative terminal of the LWB motor (left wheel backward), OUT2 pin to the negative terminal of LWF motor and positive terminal LWB motor, OUT3 pin to positive terminal of the RWF motor (right wheel forward) and negative to the RWB motor (right wheel backward). Likewise, OUT4 pin to the negative terminal of the RWF motor and positive terminal of RWB motor as exhibited in Figure 6. The complete software code of this case is presented in Appendix A.

#### Results and Discussions

Figure 7 shows the flow diagram to control the robot car with an Android-based mobile application. In this scenario, the robot car will move with the built-in Android application with different methods like touch button control (arrow and simple buttons), voice recognition. Firstly, with the touch buttons; after establishing the connection to the Android mobile application with Arduino through the Bluetooth module, whenever the user presses any of the touch buttons in the application, the corresponding signal is sent to the Arduino Uno. After receiving the signal, Arduino will check this with predefined instruction for moving forward, backward, left, right, and brake, then send the command to the motor module to move the robot in the corresponding direction.

The second method to control the robot car is voice recognition through the GUI of the android application. When the user selects the voice recognition option in the application and speaks the specific word (left, right, forward, backward and brake), the application sends this voice command signal to Arduino, and it will begin measuring the signal and complete the operation as it is defined in the previous method to control the robot car with touch buttons.

Figure 8 shows the result diagrams of the proposed robot car system controlled via the touch buttons of a Bluetooth android mobile application using Arduino. In this way, Figure 8a represents the complete working interfaces, where Figure 8a i is the starting screen of the mobile application; Figure 8a ii presents different controlling interfaces like touch arrow buttons, accelerometer, simple touch buttons, and voice recognition etc. Figure 8a iii shows the four touch arrow buttons; and Figure 8a iv illustrates the six simple buttons (functionality for each button is predefined such as, 1 for forward, 2 for backward, 3 for left, 4 for the right, 5 for brake and six is undefined). 

Although, there are five different options to control the car, here, as an example, we used touch-buttons option to demonstrate the working functionality of the proposed system. In Figure 8b, the robot car is moving forward after because the forward touch arrow button is touched in the mobile application. Same as in Figure 8c, the car is moving backward because the backward touch arrow button is pressed. In Figure 8d, the car is not moving and is stopped at mid-point because none of the touch arrow buttons were pressed. In Figure 8e, the right touch arrow button is touched in the mobile application and the robot car is turned to right direction. Similarly, in Figure 8f, the robot car moves in the left direction after pressing the left arrow button.

Furthermore, mobile-application for voice recognition is utilized. The voice recognition module in the system is limited mainly to the five voice commands (forward, stop, backward, turn left, turn right). This application converts the voice command to text and sends it to the robot via Bluetooth. The text is sent to the microcontroller from the mobile phone and it will be compared with the preprogrammed command such as left, right, etc. The microcontroller gives the commanding signal to both the wheels through the motor driver when the user gives the left command, then it stops the left wheel and the right wheel will move, moving the robot in the right direction. This android application uses google speech [86] to text to recognize and process a human voice. The method to operate the voice-controlled system is as follows: Firstly, download and install the app on the mobile phone. Turn on the mobile’s Bluetooth and run the application. Here, make sure that the robot-car is switched on, select connect to the HC-05 to authenticate the pairing (pairing password is 1234 or 0000). Finally, click on the microphone symbol on the voice recognition interface and give the voice commands. The given command will convert it into text. As the phone is connected to the microcontroller using a Bluetooth module. After the conversation of the voice command into the text, the app will send the necessary data to the microcontroller using the Bluetooth of the phone. According to the command, the robot will move forward, backward, left, right or stop. 

Figure 9 shows the result diagrams of the proposed robot car system that are controlled via voice recognition through a Bluetooth mobile-application. In this process, Figure 9a represents the voice recognition interface where Figure 9a i is the primary interface for voice recognition, and Figure 9a ii shows the interface to input voice after pressing the voice button. 

In Figure 9b, the robot car moves in a forward direction after voice recognition because the user inputs the forward move voice command to the mobile application. Similarly, the car is moving in a backward direction after voice recognition because the user inputs the backward move voice command as seen in Figure 9c. Meanwhile in Figure 9d, the user inputs the brake voice command to the application, the robot car is not moving. On the other hand, in Figure 9e, the robot car is moving in the right direction after voice recognition because the user inputs the right move voice command. Similarly, the robot car is moving in the left direction after voice recognition because the user inputs the left move voice command, which can be viewed in Figure 9f. 

We used the Arduino-based control system with wireless connectivity, in which the robot car could be automatically controlled based on hand gesture recognition and Android application (touch buttons and voice recognition), to avoid any limitations of the specifications, wireless range, etc. In some cases, we required wireless connectivity (WIFI module) to make the system more scalable for the easy integration of new devices. The Wi-Fi module could be replaced with the Bluetooth module to extend the wireless range for this system further. Moreover, the reported systems are presented as lab-scale prototypes, as aforementioned; therefore, the obstacle detection feature can be implemented for the real-scale facilities to improve safety measurements. Similarly, the proposed systems can be managed for real-scale facilities by applying other technologies like a camera [87].

It is worth noting we should consider the cases when the robot car collides with any obstacle in front of it while moving with hand gestures or mobile-application. Thus, for improving the safety measurements, we proposed a system to avoid the robot car from a collision with an obstacle by the help of a sensor (the car will stop 20 cm before the obstacle) as illustrated in Figure 10. In Figure 10a, the car is not moving to the forward direction whereas the hand is bent in the downward direction and Arduino recognized it as a forward command. Similarly, the car is not moving in a forward direction although the user pressed the forward arrow button in a mobile-app because the sensor senses that there is an obstacle in front of the car, as illustrated in Figure 10b.

The effectiveness of the proposed gesture-recognition technology will routinely add yet another dimension to human communications with consumer electronic appliances such as computers, media tablets and smartphones, video-game consoles, Microsoft’s Xbox, and PlayStation etc. The proposed gesture control and mobile-application technique is much more effective for different purposes; in the medical field it can be used by disabled patients to send messages by making a subtle gesture. It can also control electronic devices such as LEDs, fans, TVs, and air conditioners. It can help by giving feedback to augmented reality devices. It can also be paired with a smartphone to provide a more interactive and effective platform to control and interact with it, and it can also be used to control a modern electric vehicle via pairing it to the master control of that vehicle using Human Machine-Interface (HMI) technology.

## 4. Conclusions

In this paper, a design scheme for a single-equipment with multiple-application system for controlling a robot car based on Arduino has been demonstrated, which can be programmed to react to events (based on hand gesture recognition and an android mobile application as described above) and to cause corresponding actions. The proposed system is presented with principally two operational modes in which the first system uses a mechanism for controlling the robot car based on hand gesture recognition (car moves similarly to the direction and position of hand). This system is further expanded upon to compose the second mode that controls the robot car based on a mobile-application with touch buttons and voice recognition. The proposed systems have the capability to detect obstacles in front of the car. The hardware implementations of the introduced systems were provided at a lab-scale prototype to prove the simplicity, adaptability, dependability, specificity, and real low-cost of the system. We affirm that the introduced systems could be easily implemented under real requirements at large-scale in the future, and they can be efficiently implemented in smart cars, personal home robots, articulated robots, parallel robots, humanoid robots, quadrupedal robots, and hybrid robots etc.

Meanwhile, the proposed single-equipment with a multiple-application system has advantages because it is user-friendly, low-cost, low-power consumption, simple and easy to use, and the system is smaller in size. Thus a small space is required for it to adjust to the hardware circuits. In addition, the designed prototype is highly robust against unexpected problems, and can be easily extended further in the hardware section and various applications can be added to reduce the human effort of upgrading. Similarly, voice commands are sent and received through wireless serial transmission with the help of Bluetooth technology. On the other hand, as the range of Bluetooth technology is up to 10–15 m, the distance of processing the proposed system is smaller. The delay in transmission and response commands becomes high if the Bluetooth connection gets dropped frequently. Moreover, if there is noise or some other sound in the surroundings, the number of errors will increase in the proposed voice-recognition scheme. A limited number of gestures are presented here, but the algorithm can be extended in several ways to recognize a more comprehensive set of gestures. 

## 5. Future Work

Future work will build upon the improvement of the recognition system in order to increase accuracy and more gesture methods. GPS can be added to the proposed automated system by the help of which its location can be tracked. A gesture-controlled system can be controlled by head motion, eye blink, and voice-activated power. In addition, the power supply provided to the receiver can be provided through a solar power system as supply provided by solar power is in the form of a direct current (DC) and a receiver also works on DC. It can be implemented with other robot controllers without significant change. By utilizing Wi-Fi, the processing range can be extended by installing routers on short distances and by using a GSM module for wireless transmission. An onboard wireless camera can be installed, which will provide live streaming and can be used for controlling the car from faraway places. The accuracy of this system for dynamic automobiles can be improved by assessing the driving conditions [88] using a low-cost data acquisition system [89].

## Figures and Tables

**Figure 1 sensors-19-00662-f001:**
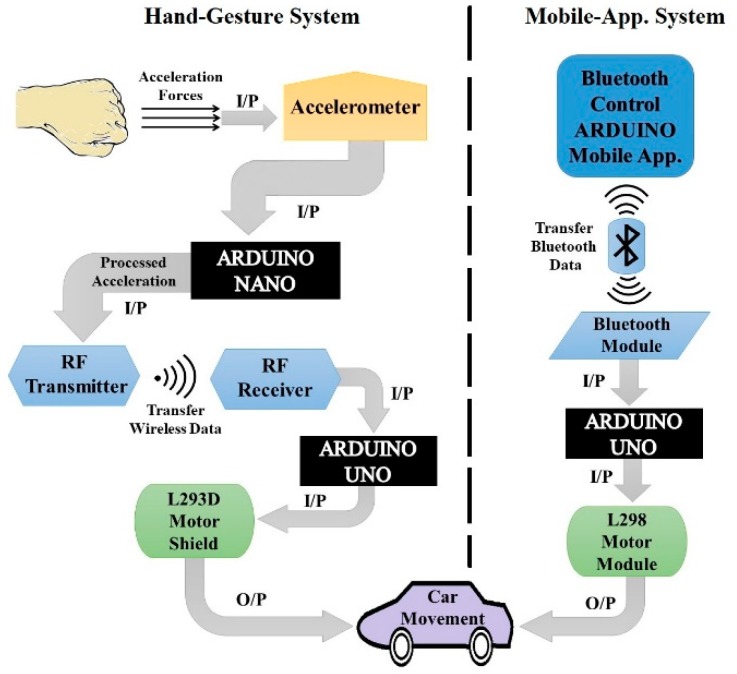
The architecture design of car controlling with hand-gesture and mobile application system.

**Figure 2 sensors-19-00662-f002:**
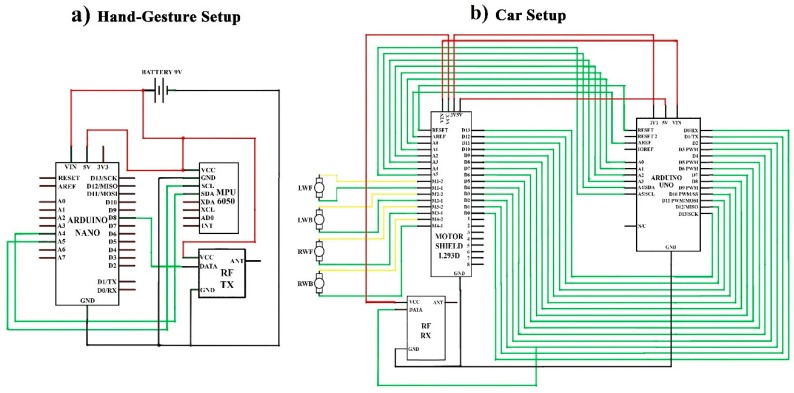
Circuit design of the car controlling system using hand gestures recognition. (**a**) Shows the schematic of the hand-gesture setup. (**b**) Displays the schematic design of car setup.

**Figure 3 sensors-19-00662-f003:**
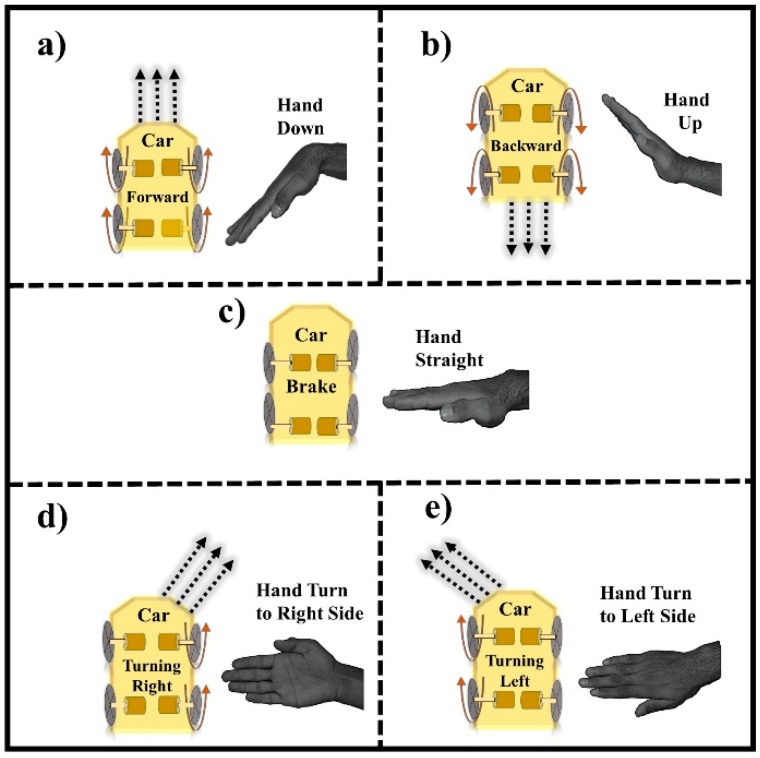
The theme of gesture recognition (i.e., movement of hand, motors, and wheels): (**a**) Hand Down; (**b**) Hand Up; (**c**) Hand Straight; (**d**) Hand Turn to Right Side; (**e**) Hand Turn to Left Side.

**Figure 4 sensors-19-00662-f004:**
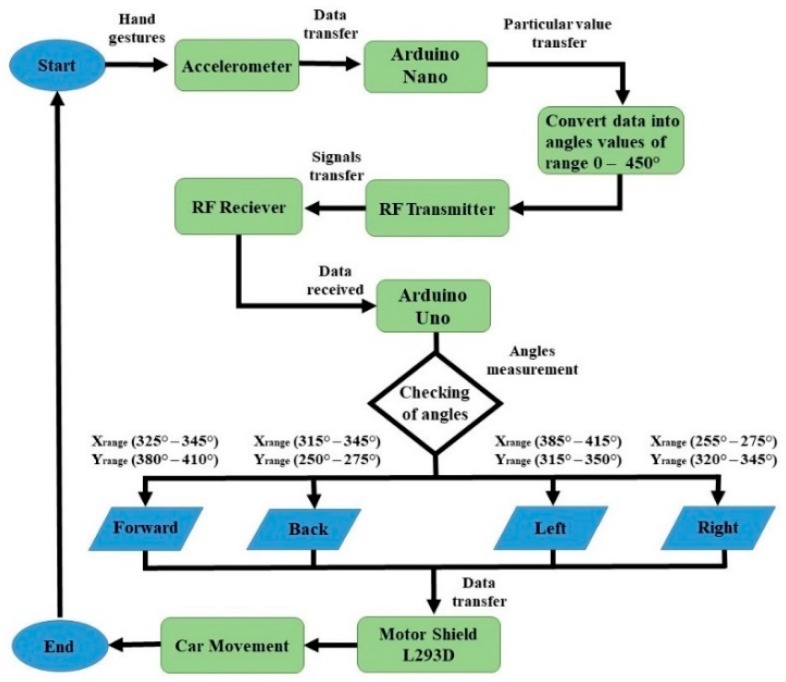
The flow diagram of the hand-gesture based car controlling.

**Figure 5 sensors-19-00662-f005:**
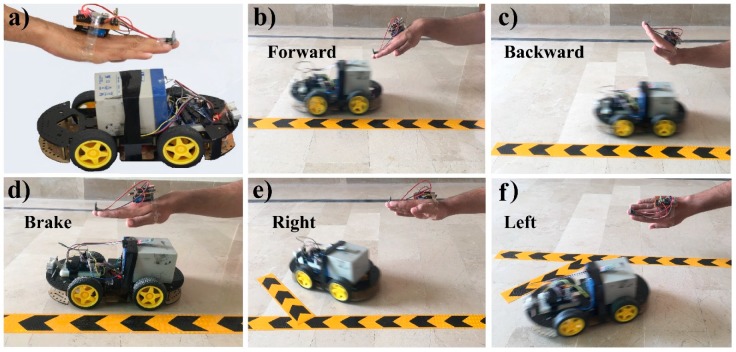
Result diagrams of the automatic robot car controlling system using hand-gestures. (**a**) Hand gesture and car’s hardware are illustrated (**b**) The hand is tilted in a downward position, so the car is moving forward. (**c**) The hand is tilted in the upward position, so the car is moving backward. (**d**) The car stopped when the hand is in a straight position. (**e**) The car is moving to right direction as the hand-tilted to the right side. (**f**) The hand is tilted to the left side, so the car is moving to left direction. The complete demonstration video of the proposed system is available online [85].

**Figure 6 sensors-19-00662-f006:**
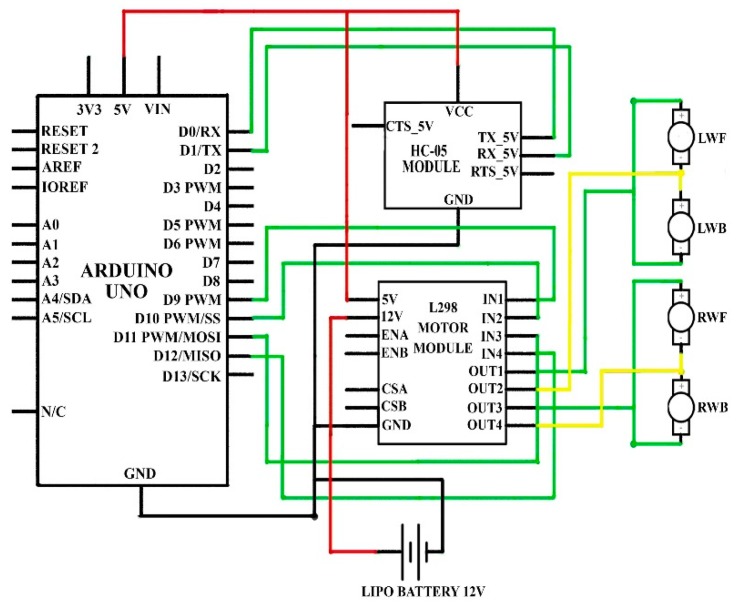
Circuit design of mobile-application system to control car.

**Figure 7 sensors-19-00662-f007:**
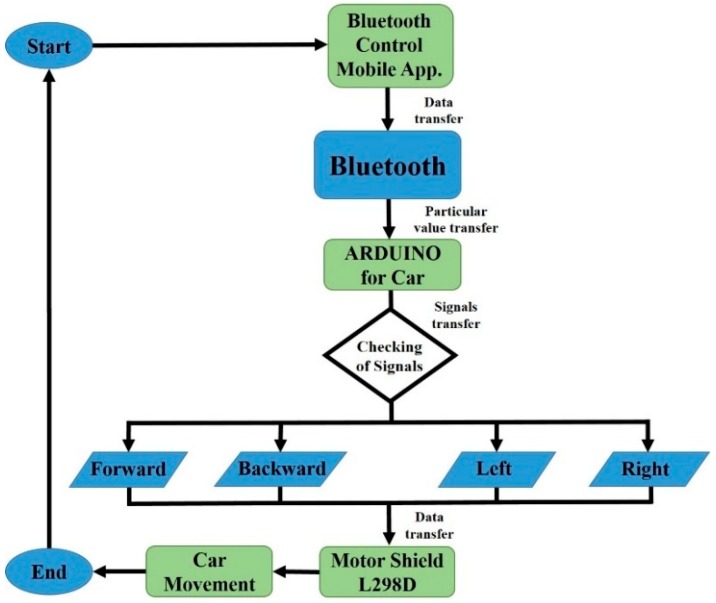
The flow diagram to control robot car with mobile-application.

**Figure 8 sensors-19-00662-f008:**
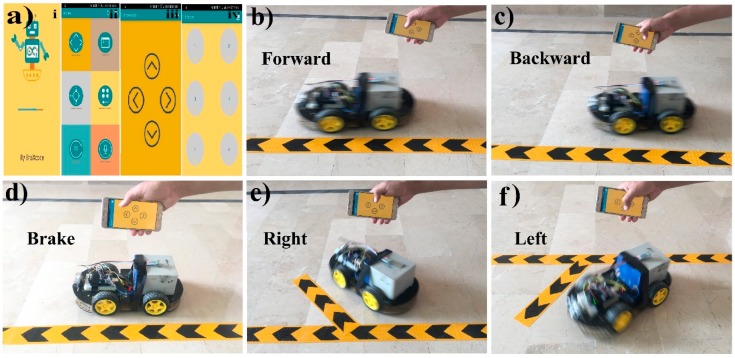
Result diagrams of the automatic robot car controlled with touch buttons of mobile-application. (**a**) i. Starting interface of the android application. ii. The homepage of an android application showing six different options of car control. iii. Touch arrow buttons interface for controlling car. iv. Simple touch buttons interface for controlling car. (**b**) The forward arrow button is touched, so the car is moving forward. (**c**) The backward arrow button is touched, so the car reverses. (**d**) The car is not moving (stopped) because none of the arrow buttons were touched. (**e**) The car is moving to the right direction as the right arrow button was touched. (**f**) The left arrow button was touched, so the car moves to the left direction. The complete demonstration video of the proposed system is available online [85].

**Figure 9 sensors-19-00662-f009:**
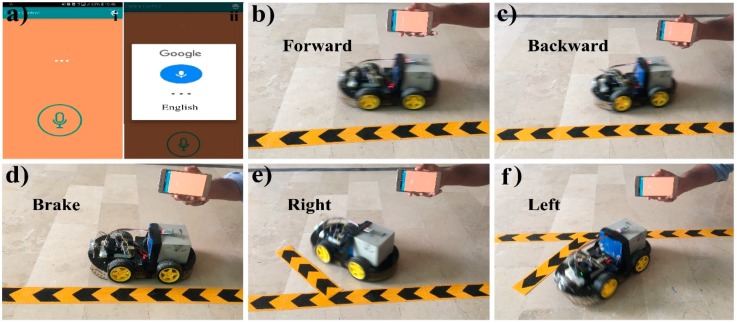
Result diagrams of the automatic robot car controlled with voice recognition in the mobile- application. (**a**) i. Starting interface of the voice recognition android application. ii. Voice is recognized after pressing the voice button of the android application. (**b**) The voice is recognized as a forward command, so the car is moving forward. (**c**) The voice is identified as a backward command, so the car is moving reversely. (**d**) The car is not moving (stopped) because the voice is recognized as a brake. (**e**) The car is moving to the right direction as the voice is identified as a right command. (**f**) The voice is recognized as a left command, so the car is moving to the left direction. The complete demonstration video of the proposed system is available online [85].

**Figure 10 sensors-19-00662-f010:**
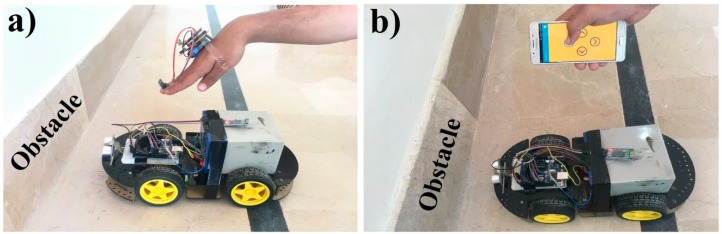
Avoidance of robot car from obstacles using hand-gestures (**a**) and mobile-application (**b**).

**Table 1 sensors-19-00662-t001:** Specification of electronic components used in to design the proposed system.

Components	Specifications
Arduino UNO [50,78]	28 pins; Operating voltage: 7–12 V
Arduino Nano [79]	30 pins; Operating voltage: 7–12 V
MPU6050 Accelerometer [80]	8 pins; Operating voltage 3.3 V
RF Sender/Receiver [81]	Sender (3 pins; Operating voltage 3–12 V; Transmission range: 90 m), Receiver (4 pins, Operating voltage 5 VDC)
L293D Motor Shield [82]	Supply-Voltage Range: 4.5–36 V; Output current: 600 mA/channel
Bluetooth Module HC-05 [83]	6 pins; Operating voltage: 3.3–5 V; Transmission range: 100 m
Android Mobile Application [77]	Android compatible
L298 Motor Module [84]	Operating voltage: 5 V; Max power: 25 W

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
