# Peer review of "Single-Equipment with Multiple-Application for an Automated Robot-Car Control System"

_sensors, 2019, doi:10.3390/s19030662_

Round 1
Reviewer 1 Report
This is the corrected version of the article: sensors-429340.
My old comments below:
The article is at a decent level and topic is interesting. Some aspects should be improved. So I can recommend to accept paper after major revision.
In order to improve manuscript, I suggest the following recommendations:
R1: Please write clearly in the "Abstract" about: 1) background (why this topic is important), 2) used methods, 3) results (present and summarize the obtained result).
R2: In the end of "Introduction" section, please write clearly: 1) list of innovative elements, and 2) identify motivations for undertaking research.
R3: In the "Introduction" section, please mention missed works about gesture recognition eg. "Hand Body Language Gesture Recognition Based on Signals From Specialized Glove and Machine Learning Algorithms", "Person Recognition based on Touch Screen Gestures using Computational Intelligence Methods".
R4: Please, improve the quality of the figures.
R5: Please specify the effectiveness of the proposed systems (controlled with hand-gestures and controlled by an android based mobile application).
R6: Please compare the proposed system with other systems described in the literature for controlling cars.
R7: Please describe more specific the voice recognition used.
R8: Please in the conclusion section: 1) list the advantages and disadvantages of the proposed solution, 2) indicate the limitations of work, and 3) indicate directions for further research.
In new version, the authors have included most of my comments so I can recommend to accept paper as it is.
Author Response
Thursday, January 31, 2019
Dear Editor,
The authors would like to thank the Editor and the anonymous reviewer for his insightful comments and constructive suggestions that certainly improved the quality of this paper. We have also mentioned this in the acknowledgement of submitted manuscript. Below please find our point-by-point response to reviewer’s concerns for article # sensors-438734.
Response to Reviewer # 01
This is the corrected version of the article: sensors-429340.
My old comments below:
The article is at a decent level and topic is interesting. Some aspects should be improved. So I can recommend to accept paper after major revision.
In order to improve manuscript, I suggest the following recommendations:
R1: Please write clearly in the "Abstract" about: 1) background (why this topic is important), 2) used methods, 3) results (present and summarize the obtained result).
R2: In the end of "Introduction" section, please write clearly: 1) list of innovative elements, and 2) identify motivations for undertaking research.
R3: In the "Introduction" section, please mention missed works about gesture recognition eg. "Hand Body Language Gesture Recognition Based on Signals From Specialized Glove and Machine Learning Algorithms", "Person Recognition based on Touch Screen Gestures using Computational Intelligence Methods".
R4: Please, improve the quality of the figures.
R5: Please specify the effectiveness of the proposed systems (controlled with hand-gestures and controlled by an android based mobile application).
R6: Please compare the proposed system with other systems described in the literature for controlling cars.
R7: Please describe more specific the voice recognition used.
R8: Please in the conclusion section: 1) list the advantages and disadvantages of the proposed solution, 2) indicate the limitations of work, and 3) indicate directions for further research.
General Comments of Reviewer: In new version, the authors have included most of my comments so I can recommend to accept paper as it is.
Reply: We appreciate the positive feedback from the reviewer. We believe that due to this reviewer’s fruitful suggestions and comments, our work became suitable for publication in well-reputed journal “Sensors”.

Reviewer 2 Report
The paper shows the development of an Arduino based controller that can be operated by gestures or by an Android application. In general the paper is well structured and writen and the novelties are clearly presented.
The title refers to a robot-car control sytem, but there is almost no literature study about car controllers or systems that have been implemented or designed specifically for cars. Therefore, although the authors suggest that this system can further been implemented in movile cars, there is no study to support this. I would recommend to add some references to address this topic for completeness.
E.g.
* Krotak, Tomas, and Martina Simlova. "The analysis of the acceleration of the vehicle for assessing the condition of the driver." Intelligent Vehicles Symposium (IV), 2012 IEEE. IEEE, 2012.
* González, A., Olazagoitia, J. L., & Vinolas, J. (2018). A Low-Cost Data Acquisition System for Automobile Dynamics Applications. Sensors, 18(2), 366.
* Cervantes-Villanueva, Javier, et al. "Vehicle maneuver detection with accelerometer-based classification." Sensors 16.10 (2016): 1618.
In the same line, it should be commented the refreshment rate that the developped Arduino-Android system is obtaining to have an idea of safety and control readiness that the system has.
Author Response
Thursday, January 31, 2019
Dear Editor,
The authors would like to thank the Editor and the anonymous reviewer for his insightful comments and constructive suggestions that certainly improved the quality of this paper. We have also mentioned this in the acknowledgement of submitted manuscript. Below please find our point-by-point response to reviewer’s concerns for article # sensors-438734.
Response to Reviewer # 02
General Comments of Reviewer: The paper shows the development of an Arduino based controller that can be operated by gestures or by an Android application. In general the paper is well structured and writen and the novelties are clearly presented.
The title refers to a robot-car control sytem, but there is almost no literature study about car controllers or systems that have been implemented or designed specifically for cars. Therefore, although the authors suggest that this system can further been implemented in movile cars, there is no study to support this. I would recommend to add some references to address this topic for completeness.
E.g.
* Krotak, Tomas, and Martina Simlova. "The analysis of the acceleration of the vehicle for assessing the condition of the driver." Intelligent Vehicles Symposium (IV), 2012 IEEE. IEEE, 2012.
* González, A., Olazagoitia, J. L., & Vinolas, J. (2018). A Low-Cost Data Acquisition System for Automobile Dynamics Applications. Sensors, 18(2), 366.
* Cervantes-Villanueva, Javier, et al. "Vehicle maneuver detection with accelerometer-based classification." Sensors 16.10 (2016): 1618.
In the same line, it should be commented the refreshment rate that the developped Arduino-Android system is obtaining to have an idea of safety and control readiness that the system has.
Reply: We appreciate the positive feedback from the reviewer. The correction has been done. We have cited the references as [15, 88, 89].

Reviewer 3 Report
Review of “Single-Equipment with Multiple-Application for an Automated Robot-Car Control System”
The paper presents an Arduino based car-controlling system which no longer require manual controlling of the cars. It is an effort to design car-controlling system that allows users to instruct car just showing it what it should do and for non-expert users, controlling the car with a remote or a switch.
The authors should analyze more the gesture recognition method with references in several techniques. There is no references on computer vision techniques. The authors should mention this kind of gesture recognition with specific examples to explain why theirs method is better.
At the beginning of section 2 (Materials and methods) the use of angle between 0° – 450° is not clear. What does mean angle 450° for the hand rotation? It must be explained.
In the paper are too much technical details. Are this details necessary for the understanding of the innovation?
The authors should check the proposed methodology – system with real data to
detect the accuracy and the efficiency
The presentation of the paper is more than adequate for publication and the topic is relevant to the journal’s aims.
The quality of the references used are of high quality.
My proposal to the editor is to accept the paper with minor corrections.
Author Response
Thursday, January 31, 2019
Dear Editor,
The authors would like to thank the Editor and the anonymous reviewer for his insightful comments and constructive suggestions that certainly improved the quality of this paper. We have also mentioned this in the acknowledgement of submitted manuscript. Below please find our point-by-point response to reviewer’s concerns for article # sensors-438734.
Response to Reviewer # 03
General Comments of Reviewer: Review of “Single-Equipment with Multiple-Application for an Automated Robot-Car Control System”
The paper presents an Arduino based car-controlling system which no longer require manual controlling of the cars. It is an effort to design car-controlling system that allows users to instruct car just showing it what it should do and for non-expert users, controlling the car with a remote or a switch.
The presentation of the paper is more than adequate for publication and the topic is relevant to the journal’s aims.
The quality of the references used are of high quality.
My proposal to the editor is to accept the paper with minor corrections.
Reply: We appreciate the positive feedback from the reviewer.
Comment #1: The authors should analyze more the gesture recognition method with references in several techniques. There is no references on computer vision techniques. The authors should mention this kind of gesture recognition with specific examples to explain why theirs method is better.
Reply #1: References for computer vision techniques are mentioned [56,57].
The computer vision system is subject to a limited range of view. A camera is fixed in a specific place while tracking the moving object. It is challenging to perform image processing using one camera to identify an object and to find its position (x,y). Most of the computer vision system is developed with the Calibration of using MATLAB built-in function and also uses a unique algorithm that scans both images to find a match feature. The problem in the vision-based system is to guide the robot such that the tracked trajectory must be equal to the desired trajectory.
Our system is better because it cannot use a computer or a unique algorithm for vision recognition, nor uses special software like MATLAB.
Comment #2: At the beginning of section 2 (Materials and methods) the use of angle between 0° – 450° is not clear. What does mean angle 450° for the hand rotation? It must be explained.
Reply #2: Mpu6050 [ref. 80] is a Tri-Axis angular rate sensor (gyro) with a sensitivity up to 131 LSBs/dps and having a pre-defined full-scale range of ±250, ±500, ±1000, and ±2000dps. So, we select only the unique set of ranges from 0 – 450° accordingly to the direction of hand palm for our system.
Comment #3: In the paper are too much technical details. Are this details necessary for the understanding of the innovation?
Reply #3: We believe that the detail is necessary for the understanding of the innovation. We proposed our system with full technical information to quickly implement this system even by non-experts.
Comment #4: The authors should check the proposed methodology – system with real data to
detect the accuracy and the efficiency.
Reply #4: The system can efficiently deal with real data to detect accuracy and efficiency because it uses simple and easily available tools that work perfectly in the real world. The methods are proposed with full detail that can be implemented effortlessly even by a non-expert person.

This manuscript is a resubmission of an earlier submission. The following is a list of the peer review reports and author responses from that submission.
Round 1
Reviewer 1 Report
The article is at a decent level and topic is interesting. Some aspects should be improved. So I can recommend to accept paper after major revision.
In order to improve manuscript, I suggest the following recommendations:
R1: Please write clearly in the "Abstract" about: 1) background (why this topic is important), 2) used methods, 3) results (present and summarize the obtained result).
R2: In the end of "Introduction" section, please write clearly: 1) list of innovative elements, and 2) identify motivations for undertaking research.
R3: In the "Introduction" section, please mention missed works about gesture recognition eg. "Hand Body Language Gesture Recognition Based on Signals From Specialized Glove and Machine Learning Algorithms", "Person Recognition based on Touch Screen Gestures using Computational Intelligence Methods".
R4: Please, improve the quality of the figures.
R5: Please specify the effectiveness of the proposed systems (controlled with hand-gestures and controlled by an android based mobile application).
R6: Please compare the proposed system with other systems described in the literature for controlling cars.
R7: Please describe more specific the voice recognition used.
R8: Please in the conclusion section: 1) list the advantages and disadvantages of the proposed solution, 2) indicate the limitations of work, and 3) indicate directions for further research.
Reviewer 2 Report
The paper proposed a multi-modal (3D hand gesture, voice, and touch button on android app) car controller system via arduino board. The authors have done a good job in their presentation.
However, the paper itself lacks of research contribution to be considered for the research journal. It is not clear what is the research challenges that they faces and how to solve further than combining different existing modules together to build the system.
The paper needs to focus more on that aspect and de-emphasizes details that can be found elsewhere (book, internet, etc.). such as Fig. 2, 6, section 2.